# Improving Cohort-Hospital Matching Accuracy through Standardization and Validation of Participant Identifiable Information

**DOI:** 10.3390/children9121916

**Published:** 2022-12-07

**Authors:** Yanhong Jessika Hu, Anna Fedyukova, Jing Wang, Joanne M. Said, Niranjan Thomas, Elizabeth Noble, Jeanie L. Y. Cheong, Bill Karanatsios, Sharon Goldfeld, Melissa Wake

**Affiliations:** 1Murdoch Children’s Research Institute, The Royal Children’s Hospital, Parkville, VIC 3052, Australia; 2Department of Pediatrics, The University of Melbourne, Parkville, VIC 3052, Australia; 3Department of Obstetrics and Gynaecology, The University of Melbourne, Parkville, VIC 3010, Australia; 4Maternal Fetal Medicine, Joan Kirner Women’s & Children’s at Sunshine Hospital, St Albans, VIC 3021, Australia; 5Newborn Research, The Royal Women’s Hospital, Parkville, VIC 3052, Australia; 6Western Health Chronic Disease Alliance, Western Health, St Albans, VIC 3021, Australia; 7Centre for Community Child Health, The Royal Children’s Hospital, Parkville, VIC 3052, Australia

**Keywords:** birth cohort, hospital, data linkage, pregnant women, newborn, hospital records, information retrieval, personally identifiable information, data accuracy, demographics

## Abstract

Linking very large, consented birth cohorts to birthing hospitals clinical data could elucidate the lifecourse outcomes of health care and exposures during the pregnancy, birth and newborn periods. Unfortunately, cohort personally identifiable information (PII) often does not include unique identifier numbers, presenting matching challenges. To develop optimized cohort matching to birthing hospital clinical records, this pilot drew on a one-year (December 2020–December 2021) cohort for a single Australian birthing hospital participating in the whole-of-state Generation Victoria (GenV) study. For 1819 consented mother-baby pairs and 58 additional babies (whose mothers were not themselves participating), we tested the accuracy and effort of various approaches to matching. We selected demographic variables drawn from names, DOB, sex, telephone, address (and birth order for multiple births). After variable standardization and validation, accuracy rose from 10% to 99% using a deterministic-rule-based approach in 10 steps. Using cohort-specific modifications of the Australian Statistical Linkage Key (SLK-581), it took only 3 steps to reach 97% (SLK-5881) and 98% (SLK-5881.1) accuracy. We conclude that our SLK-5881 process could safely and efficiently achieve high accuracy at the population level for future birth cohort-birth hospital matching in the absence of unique identifier numbers.

## 1. Introduction

Health services research using de-identified methods to link administrative datasets, typically undertaken with ethical approval for a waiver of consent, can generate powerful population insights to help shape care and policy [1,2]. Such linkage often uses a unique national administrative identifier, such as Scandinavian countries’ unique personal identity number [3], US social security number [1], UK National Health Service (NHS) and National Insurance numbers [4], and Australia’s national health insurance number (Medicare) [5]. Because linkage via these unique identifiers is highly accurate, it tends to dominate the published data linkage literature [5,6,7].

We propose that more, and more impactful, health services research could be undertaken if very large, consented birth cohorts were able to readily link to clinical data, including from hospitals. Firstly, such cohorts typically collect molecular, phenotypic, and participant-reported data that administrative datasets lack. Secondly, hospitals clinical data may be especially important for the pregnancy and newborn periods, as (1) these are the only life points at which hospital care is essentially universal in many countries [3,6,8,9,10]; (2) there is substantial variation in care [11]; and (3) these periods shape subsequent outcomes for the entire maternal and child life course [12]. By accessing deep clinical data, consented cohorts could fill many gaps in both predicting health care and understanding its impacts on health and wellbeing [13]. However, other than minimum administrative datasets, most elements of Australian hospital care are neither available for research nor in a standardized format; for example, hospitals vary widely in how they record prescriptions and pathology results [5] and whether/how they collate them. Thus, data extraction may need to be on a hospital-by-hospital basis with subsequent standardization via common data models.

High matching rates between cohorts and hospital records have been achieved when unique identifiers are available, for example true matching rates above 97% for a UK study [14] and nearly 100% for a Hong Kong study [10]. Unfortunately, many consented cohorts do not hold hospital unique identifiers; participants may not know them, and it may not be easily possible to obtain them from services due to privacy and legal considerations [4,5]. This poses challenges for accurate matching, which then relies on participants’ details such as name, postcode and date of birth. Such non-unique identifiers can facilitate linkage but also lead to linkage error and uncertainty [15]. While large-scale anonymized analyses may absorb some uncertain matches or mismatches, higher accuracy may be needed when consented clinical data permanently enter a major dataset designed to address future research questions, especially if return of results is part of the cohort’s framework. Consequently, even a small improvement in matching accuracy could improve the health services research utility of population-based cohorts.

However, it is not clear how best to achieve high matching rates for cohort studies in the absence of a hospital unique identifier. Demographic identifiers vary in format and details both among cohorts and among hospitals. In Australia, the GRHANITE^TM^ Linkage Tool reached high sensitivity (95–100%) for clinical and pathology service data for linkage including Medicare number, but sensitivity dropped to only 66% in the absence of Medicare number [16]. The Australian Institute of Health and Welfare has created Statistical Linkage Key (SLK-581) to link hospital and death records through a probabilistic strategy, achieving matching rates of 97.5% [17]. However, this strategy may not be suited to deterministic matching, matching at an individual level, or where names differ significantly between two datasets (which is more likely at the start than the end of life) or large missing values [18].

We address this gap with the Generation Victoria (GenV) cohort, a statewide population-based birth cohort now open to all newborn babies and their mothers over two years in all 58 birthing hospitals in the state of Victoria (population 6.5 million), Australia [19]. Consent includes permission to bring into GenV information and samples that services already collect in clinical practice. In a one-year GenV sub-cohort from a single birthing hospital, this pilot study aimed to (1) compare different approaches to improve patients-participants matching accuracy after standardization and validation, and (2) recommend an approach for statewide scale up.

## 2. Materials and Methods

We conducted this data linkage matching report in accordance with reporting guidelines for studies involving data linkage [20] and in line with REporting of studies Conducted using Observational Routinely-collected Data (RECORD) guideline [21] (see Appendix A). The first step was for the hospital to undertake initial selection of patients who appeared to also be GenV participants; for this, we tested three scenarios regarding how the hospital could best identify the possible probands. Once the hospital returned the initial potentially matched datasets, the second step was for GenV to then undertake further standardization/validation followed by optimization (highest accuracy, lowest effort of matching) comparing three different matching approaches. GenV data scientist and hospital analyst both had the authorization of data access, separation principle has implemented for PII and clinical data to protect patients privacy.

### 2.1. Sampling Frame and Recruitment

*GenV cohort participants*: Participants for this study were recruited to GenV [22] from a single birthing hospital from commencement of recruitment on 5 December 2020 to 31 December 2021. The study includes all mother-baby pairs in this period, plus consented babies whose birth mothers were not themselves participating in GenV (see Table 1 for inclusion criteria).

GenV’s sampling frame is the daily census of all births at each birthing hospital, plus notification of any in-transferred newborns not recruited at the birthing hospital. Hospital-based GenV-employed recruiters attempt to approach the parent/s of every baby face to face before discharge. They seek parent consent to follow participants indefinitely until study end or withdrawal, with a primary parent/guardian asked to provide consent for themselves and their child (index participant) and any additional parents/guardians asked to consent for themselves only. The broad consent includes to GenV accessing the information that services already collect for them and their child, from before the baby was born and in the future. The recruiter collects parent and child demographic details from the parent and records them in GenV’s Participant Relationship Management System (PRMS). At the time of recruitment for this pilot, this did not include recording the unique hospital identifier the Unit Record (UR) Number, a permanent identifier that is assigned to the patient and comprises a digitized number and/or letter combination unique to individuals in an Australian health service [23].

The birthing hospital: The participating birthing hospital has around 5800 births annually and covers much of the culturally diverse western suburbs of Melbourne, Australia. The sampling frame for the hospital dataset spanned an additional month each side of the GenV sample (i.e., November 2020 to January 2022) to ensure data completeness.

### 2.2. Data Sources and Handling

This study required that we match participants on demographic details (personally identifiable information, PII) in the single GenV PRMS dataset with those that hospital held across two data systems. It further required that the hospital identified which individual babies belonged to which individual mothers. We explored matching using variables available in GenV (Table 2): mother and baby first name (FN), middle names, last name (LN), date of birth (DOB), birth order (BO—baby only, for multiple births), baby sex (SEX), mother/the other parent’s phone number (TN) and home address (ADD), and mother’s and baby’s computer-generated ID.

Figure 1 shows the process flows. The GenV data scientist securely extracted the participant variables from GenV’s PRMS, encrypted the data in a single CSV file saved in a secured Owncloud account, and provided the login information to hospital analyst. The hospital data analyst undertook initial matching resulting in a separate hospital dataset (Appendix A) and then returned both the original GenV personally identifiable information (PII) and the new hospital datasets to GenV Owncloud account. The GenV data scientist then undertook all subsequent steps, in communication with the GenV authorized recruitment team and the hospital analyst, until the optimized matching rate was reached.

### 2.3. Step 1 (Hospital): Initial Matching

Data linkage used two systems: Patient Administration System (iPM) [24] and the Birthing Outcomes System (BOS) [25]. iPM accumulates patients’ details (e.g., name, address, DOB, telephone) and their admission details, such as admission and separation dates and admission type. BOS is a standalone product, which stores birthing and pregnancy data (including birth order, baby sex) independently from iPM. The two systems are internally linked within the hospital by the UR number.

The hospital data analyst linked GenV participants with minimum sets of variables utilizing three scenarios (see Table 3): (1) primarily using the mother’s details to link the mother and baby, (2) primarily using the baby’s details to link the mother and baby, and (3) using all available variables for babies with no information about their mothers but the other parents’ information.

### 2.4. Step 2 (GenV): Standardization/Validation and Optimization of Matching Approaches

To improve matching rate, we implemented standardization and validation for the PII data variables of names, telephone number and address during further analysis of both the original GenV dataset and the three datasets returned from the birthing hospital after potential matches were identified.

LN and FN: As both datasets may have spelling errors, where they differed, we could not know which were the true correct names. We standardized names by removing spaces, hyphens and special symbols (‘) between two words; all letters were upper cased for both FN and LN. Although neither dataset had missing values for names of mothers or babies, this often included the non-specific ‘Baby’ in the FN field.

TN: We standardized telephone numbers in adherence to Australian Telecommunications standards (Figure 1) [26]. The hospital’s dataset has two TN variables (‘mobile number’ and ‘other phone’) while the GenV cohort has one TN variable (with 1.1% of values missing). We combined the two hospital TN variables into one TN variable and reduced the missing values to 3.5%. This required converting raw telephone numbers into the standard format and the removal of special characters such as apostrophes or hyphens. In addition, country code was removed and the leading ‘0’ was removed.

ADD: We applied Australian Postal Service certified address standardization rules for thoroughfare abbreviations (Figure 2) [27]. Hyphens, empty spaces, and special symbols were removed. For example, address variables in both datasets had the following format after pre-processing: ‘214SMITHST’ (Appendix A). In the GenV dataset 1.4% of parents’ address values were missing, but there were no missing values in the hospital dataset.

Figure 2 illustrates the example of data standardization and validation between the GenV and hospital datasets on the variables of phone number and address.

We tested three different approaches after data were standardized and validated: a deterministic rule-based approach and two modified SLK-581 approaches.
Deterministic rule-based approach: Use all mothers’ and all babies’ PII (Detail in Table 2).Modified SLK-581 approach 1—SLK-5881: SLK-581 [28] is a 14-character code comprising the 2nd, 3rd and 5th characters of the family name, the 2nd and 3rd of the given/first name (‘5’), the date of birth (DDMMYYYY, ‘8’) and sex (‘1’). This has previously provided successful linkage in some datasets but less so in studies using data with a high rate of missing names [18]. As the GenV cohort includes twin and triplet babies with imprecise names (e.g., ‘Boy’, ‘Girl’, ‘Twin1’, ‘Twin2’) and the sex of the babies may also be imprecise, we modified the SLK-581 by adding babies’ B-DOB and BO as ‘birth order’ was the only unique identifier for twins. We removed the mother’s sex variable as mother’s sex was the same in both datasets. We named this new linkage method as SLK-5881 as the ‘5’ (2nd, 3rd letter of FN, and 2nd, 3rd, 5th letter of LN), the first ‘8’ remains the mother’s DOB, with the additional ‘8’ for baby’s DOB and ‘1’ for birth order.Modified SLK-581 approach 2—SLK5881.1: Additionally, we tested a modified linkage method using the first 2 letters of first name and the first 3 letters of last name. M-DOB, B-DOB, B-DOB and BO were used as per SLK-5881.1.

### 2.5. Statistical Analysis and Evaluation Metrics

We defined three categories for matching results: fully matched, non-matched and partially matched. True matches were defined when all selected data variables were identical between GenV and the hospital datasets. Non-matches referred to when a GenV participant recruited from that hospital was not found in the hospital dataset. Partial matches referred to when one or more selected variables were matched while the rest of the variables had discrepancies, which required a further manual check to determine whether matches belong to true matches or non-matches. The matching performance quality was evaluated using Accuracy rate = (true matches)/(true matches + non-matches + partial matches) × 100%. We used manual comparison as the gold standard to confirm all true matches for this pilot.

### 2.6. Ethics, Privacy, and Data Protection

Ethical approval is in place for the GenV cohort (Royal Children’s Hospital Human Research Ethics Committee (HREC)-2019/11), including consent to access clinical data. Written informed consent was obtained from participants. For this participating hospital data linkage matching, we further obtained site-specific governance authorization, including site-specific assessment, material transfer agreement and privacy assessment before PII data extraction commencement. 

## 3. Results

### 3.1. The Sample

The GenV cohort dataset included 1819 mother-baby pairs and 58 babies without mothers’ PII recruited from the participating hospital between December 2020 and December 2021. The hospital identified a total of 22,236 adult patients (by using 1st letter of M-FN and 2nd letter of M-LN and year of M-DOB), 5565 babies with mothers’ PII (1st two letters of B-LN or M-LN and year of B-DOB) and 76 babies without mothers’ PII (1st 2 letters of baby LN and year of B-DOB) from November 2020 to January 2022 with which to undertake their initial matching.

### 3.2. Step 1: Hospital Initial Matching in the Three Linkage Scenarios Analysis

The hospital data analysist assessed the initial matching and identified the most promising scenarios for further analysis. This process determined whether we use the mothers’ or babies’ PII for matching (Figure 3) and without manual check.

In Scenario 1, using adult patients’ PII details for matching, 1919 possible mother-baby pairs were identified in the hospital dataset. After removing 69 duplicates (twins), there were 160 fully matched pairs, 1622 partially matched and 37 non-matched pairs.

In Scenario 2, using babies’ PII details for matching, 2289 possible mother-baby pairs were identified in the hospital dataset. There were 94 fully matched pairs, 1555 partially matched and 170 non-matched pairs.

In Scenario 3, for the 58 babies without mothers’ details, 53 babies from the hospital dataset were initially identified, with 28 babies partially or fully matched. A combination of other parents’ and babies’ PII details was used for further matching process.

The hospital then returned the potential matches from all three scenarios to GenV, comprising the selected PII variables of the overlapping 1919 mother-baby pairs and 2289 mother-baby pairs identified above (with 1650 pairs common to both) and 53 babies without mother’s information but with the other parents’ ADD and TN information.

### 3.3. Step 2 (GenV): Standardization/Validation and Optimization of Matching Approaches

Without data pre-processing and standardization, the accuracy rate for scenario 1 and 2 with 6 variables (M-FN, M-LN, M-DOB, B-DOB, B-BO, M-address) was 9% (160/1819) and 5% (94/1819), respectively. In Scenario 3, with 4 variables B-LN, B-DOB, Other parent’s address (ADD) and phone (TN) used for initial matching, the accuracy rate was 3% (2/53).

Appendix A describes the three different approaches and their sequential steps with different demographic variables, all true matches were confirmed manually. Figure 4 shows the three different approaches and their accuracy rates at each step (noting that subsequent recommendations related not only to accuracy but also to effort required).

#### 3.3.1. Deterministic Rule-Based Approach

##### Mother-Baby Pairs

The matching accuracy rate was 9% (168/1819) before standardization. After data standardization and validation, we proceeded to sequential matching steps. Step 1 employed 7 variables from mothers and babies (M-FN, M-LN, M-DOB, M-ADD, M-TN, B-DOB, B-BO) as linkage keys for matching, and the accuracy rate was 65% (1189/1819). In Step 2, we used 6 variables, excluding address (M-FN, M-LN, M-DOB, M-TN, B-DOB, B-BO) and the accuracy rate increased to 82% (1486/1819). In Step 3, we used 6 variables without FN, and the accuracy rate increased to 86% (1571/1819). In Step 4, we used 6 variables without mother’s TN and the accuracy rate reached 90% (1648/1819). In Step 5, we used 6 variables without M-DOB, and the accuracy rate increased to 93% (1694/1819). By step 10, the accuracy rate reached 99% (1800/1819). Appendix A shows the details of all 10 steps and the variables used for each step. Appendix A provide examples of and reasons for partial matching requiring further manual checks.

After manually checking through both the hospital and cohort datasets, we then reached 100% matching accuracy rate for mother-baby pairs.

##### Babies without Mother’s PII

In the GenV cohort dataset, 58 babies recruited from this sampling frame did not have mothers’ PII. Before standardization and validation, the matching rate was 0% as there were no matches for any of the 58 babies in the hospital dataset. After standardization and validation, it took 3 steps to match those babies. In Step 1, with baby variables of name, DOB, SEX, BO and the other parent’s TN and ADD, the accuracy rate was 3% (2/58). In Step 2, baby variables included LN, DOB, SEX, BO and other parent’s ADD; the accuracy rate was 43% (25/58). In Step 3, only baby’s FN, LN, DOB, SEX, BO were included; the accuracy rate was 48% (28/58) (see Table 3).

#### 3.3.2. SLK-5881 Matching Approach

##### Mother-Baby Pairs

Similar to the deterministic rule-based approach, we applied the SLK-5881 approach with matching steps sequentially. The matching accuracy rate was 5% (99/1819) before standardization. In step 1, accuracy rate reached 93% (1687/1819) by using 5 variables including 2nd and 3rd letters of M-FN, 2nd and 3rd and 5th letters of M-LN, M-DOB, B-DOB, B-BO. In step 2, accuracy rate achieved 96% (1755/1819) by using 5 letters of mother’s name, M-DOB, B-DOB, B-BO and M-TN. In step 3, accuracy rate reached 97% (1764/1819) (Appendix A).

##### Babies without Mother’s PII

All steps for SLK-5881 approach were sequential with 53 babies identified in hospital dataset without manual check. Before standardization and validation, the accuracy rate was 17% (10/58). After standardization and validation, we used 2nd, 3rd and 5th letters of B-LN for all steps in this approach. In step 1, we included additional 2nd and 3rd letters of B-FN, B-DOB, B-Sex and BO, the accuracy rate was 24% (13/53). In step 2, additional B-DOB, B-Sex and BO and the other parent’s M-TN were included, the accuracy rate was 26% (14/53). In step 3, additional B-DOB, B-Sex and BO and other parent’s address were used for matching, accuracy rate was 45% (26/53).

#### 3.3.3. SLK-5881.1 Matching Approach

##### Mother-Baby Pairs

Similar to SLK-5881 approach, we employed SLK-5881.1 approach for matching linkage. The matching accuracy rate was 9% (168/1819) before standardization. After standardization and validation, we achieved 94% accuracy rate (1709/1819) in step 1, 97% (1764/1819) in step 2, and 98% (1782/1819) in step 3. See Appendix A for the data variables used.

##### Babies without Mother’s PII

We adjusted SLK-5881.1 approach by using baby’s names and the other parent’s TN and ten symbols (letters and numbers) of the other parent’s ADD variable to match those babies without mothers’ PII. Like mother-baby pair matching, all steps were sequential. The accuracy rate for SLK-5881.1 was 17% (10/58) before standardization and validation. After standardization and validation, the accuracy was 26% (15/58), 29% (17/58) and 48% (28/58), respectively in step 1, step 2 and step 3 (Appendix A).

Appendix A shows reasons for partially matched pairs requiring further manual check for both SLK-5881 and 5881.1.

## 4. Discussion

### 4.1. Principal Results

High matching to birthing hospital records of mother-baby pairs is possible for a consented cohort without unique identifiers. Before standardization and validation, the accuracy rate was less than 10%. After standardization and validation, all 3 approaches showed very high success rates for the mother-baby pairs (100% vs. 97% vs. 98%). Modified SLK-581 approaches were faster and involved much less effort (3 steps) than the deterministic rule-based approach (10 steps) and should enable efficient scaling when linking the whole GenV cohort to multiple birthing hospitals across our state. For babies without mothers’ PII, none of our three standardized approaches achieved a matching accuracy rate above around 50%; direct manual perusal of hospital and cohort records did eventually achieve 100% matching, but this may not be scalable for a large number.

While some demographic variables were already largely standardized (e.g., DOB) across both data sources, standardization and validation of other demographic variables greatly improved matching performance between the GenV and hospital datasets. Babies’ first names could not be standardized as for many this was simply listed as ‘Baby’, especially in the hospital dataset.

### 4.2. Comparisons with Published Studies

For our consented cohort using only demographic summary variables, we achieved high matching rates to birthing hospitals records using efficient, secure methods. These matching rates were comparable to studies using combinations of unique identifiers (lacking in our dataset) and the child’s and mother’s demographic information, e.g., the studies from UK [14] and Hong Kong [10]. We have found no other reported studies that have achieved this.

Several studies have, however, shown that standardized variables can improve matching over and above that with unique identifier numbers alone. A study from the US (matching health information exchange (HIE) records, public health registry, Social Security Death Master File records and newborn screening records) showed standardizing individual variables (telephone and date of birth, as well as social security number) increased matching sensitivity from 81.3% to 91.6%; however, standardizing address and last name showed no improvement [29]. A Canadian study achieved 95% matching of records across community health service agencies via a weighted approach; matches based on Health Card Number and Last name were weighted 1.0, while matches based on the Last name, First name, DOB, and Gender were weighted 0.7 [30].

### 4.3. Implications

Taking Australian experience as a case study, multiple barriers face researchers attempting to bring together birth cohort and hospitals data. Because hospitals use diverse records of varying sophistication, ranging from fully electronic to handwritten medical records within and across hospitals, clinical data are not brought together in any unifying or collated way beyond the minimum Victorian Admitted Episodes Dataset (VAED), and each hospital’s unique identifier number applies only to that hospital. In this situation, cohorts must work across inconsistent health record systems on a hospital-by-hospital basis, without the benefit of a unique identifier—but, until now, there have been no reports as to how to achieve this to the high level of accuracy cohorts may require. Our study shows that it is possible to navigate the necessary privacy and legal requirements and work in a cohort-hospital partnership to this end. Further, by modifying an existing Australian linkage key (SLK-581) we provide a short set of variables (phone, addresses with additional mothers and babies’ date of birth, baby’s birth order) and steps that can be efficiently applied to achieve high rates of matching for mother-baby pairs in birthing hospitals. On the other hand, for many Australian babies are born under mothers’ name, but not all, Australian babies after birth (e.g., babies born to sole mothers, gay parents, or whose parents choose to give them the mother’s surname). Our approach has proved sufficiently flexibility to match regardless of whether babies’ surname is the same or different from their mothers’ surnames. These are likely to be generalizable to other hospitals, regardless of which health record system they are using and what the formats of those demographic data are.

Performance of the SLK-5881 and SLK-5881.1 approaches was comparable. The former might be seen as offering superior privacy protection by virtue of its use of variables in selected positions, rather than consecutive letters, which perhaps could be more identifiable in the event of data breaches.

While undertaken securely, our processes were not fully de-identified, in keeping with the consent of the cohort participants. Based on our experience, highly accurate fully de-identified linkage is unlikely without unique numbers such as unit record (UR) number, Medicare number [31] or some other individual healthcare identifier [32]. However, there is increasing concern about sharing such numbers both by individuals and from a legal perspective [33]. With many techniques proposed for privacy-preserving clinical linkage [34], we hope that future safe access to such datasets for beneficial research will be widely supported and enabled [13,35] in ways that meet the needs both of citizens and good governance.

### 4.4. Limitations

There are several limitations to this study. First, our results are specific to the hospital we selected. However, there is no reason to think that the demographic variables we used would differ greatly from those in other Australian hospitals [36]. Therefore, our findings are likely to apply for birthing hospital-to-cohort exchanges throughout Victoria and potentially nationally and to other countries with computerized administrative databases; however, we acknowledge that these may not always be available in low-middle income countries, which may raise different security concerns than managed here. Second, this approach may not apply to matching of babies without mothers’ information as the matching accuracy rate was low with all three approaches. Third, this standardization might not cover all the potential partial matches which might occur in future data sources with, for example, name changes with altered family compositions over time. However, this first experience is very promising for GenV’s ability to access hospitals data at a critical universal lifecourse juncture (the pregnancy, birth and newborn periods) and will help optimize our process as we scale up to include more hospitals.

## 5. Conclusions

Standardizing participants’ names, phone and address demographic data led to very high cohort-to-hospital matching rates both for the mothers and the babies in our large, consented birth cohort. This was achieved despite the cohort not holding hospital unique identifier numbers. We recommend our modified SLK-5881 approach as achieving the best balance of accuracy, efficiency, safety and scalability to the population level.

Because health care and health status during pregnancy, birth and the newborn periods can influence the whole lifecourse, linking to hospital health records covering these periods could provide immense additional value to cohorts in their quest to improve maternal and child health.

## Figures and Tables

**Figure 1 children-09-01916-f001:**
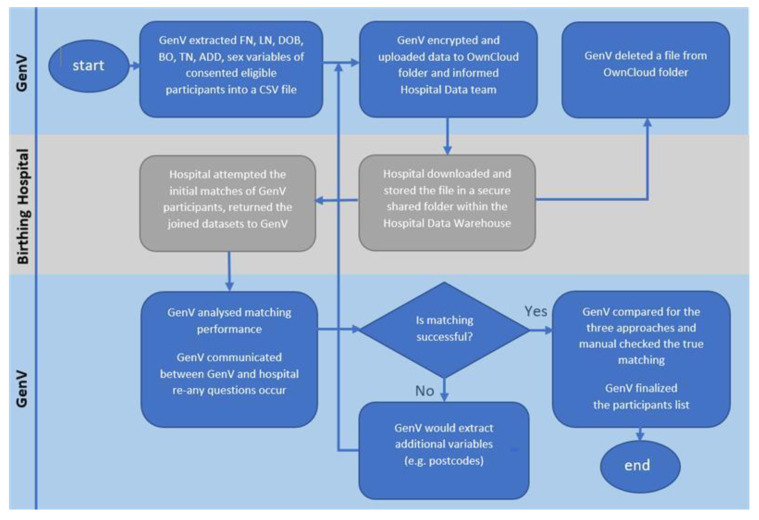
Linkage matching process flow. Note: All matching and data retrieval were done by authorized data scientist from GenV, data initial matching was done by authorized hospital data analyst. FN = first name; LN = last name; DOB = date of birth; BO = birth order; TN = telephone; ADD = addresses.

**Figure 2 children-09-01916-f002:**
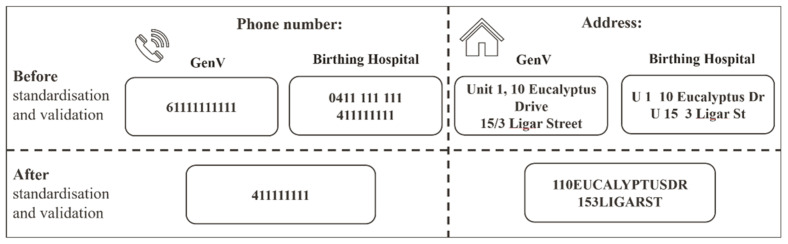
Data standardisation and validation examples for telephone and address. Note: Phone number and address are mock-up examples.

**Figure 3 children-09-01916-f003:**
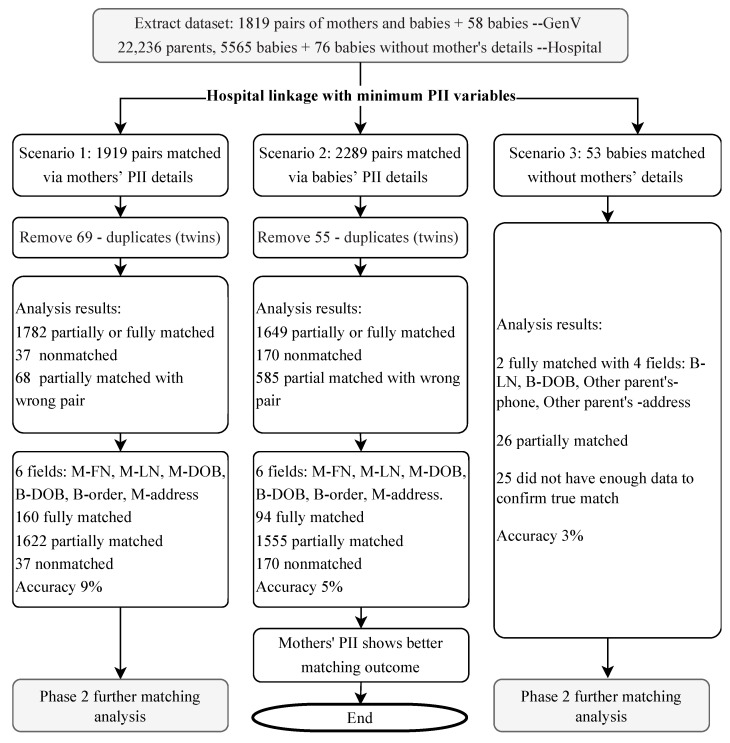
Three linkage scenarios of hospital initial matching. Note: PII = personally identifiable information.

**Figure 4 children-09-01916-f004:**
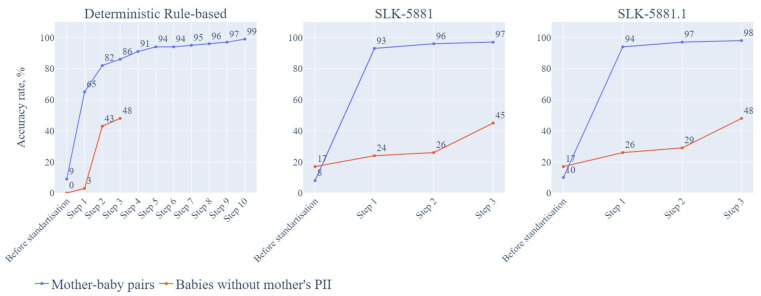
Accuracy rates for mother-baby pairs and babies without mothers’ PII information and their required steps for the three approaches. Note: details of included variables for each step listed in Appendix A. PII = personally identifiable information; SLK = Statistical Linkage key.

**Table 1 children-09-01916-t001:** Criteria for inclusion or exclusion in GenV and the linkage matching cohort.

**GenV Birthing Hospital**	
Inclusion criteria	All children born in Victoria during the recruitment period whose parents/guardians have decisional capacity, and their parents Participants who leave Victoria may continue to take part via linked and contributed data Families who move to Victoria later and have children born within the recruitment period may join GenV GenV recruitment and data collection materials are offered in multiple languages to enhance accessibility
Exclusion criteria	Infants who die before recruitment to GenV (stillbirth or neonatal death) Families unable to consent in any available language
**Linkage Matching Cohort**	
Inclusion criteria	Baby is born between December 2020 and December 2021 Consented babies and parents who agreed to participate in GenV There is a record for admission between November 2020 and January 2022 at the selected Victorian birthing hospital.
Exclusion criteria	No additional exclusion criteria

**Table 2 children-09-01916-t002:** PII data variables explored for matching from GenV mothers and babies.

**Mother or Baby**	PII Data Variables
**Both**	ID, generated by computer
**Baby**	First Name (B-FN)
	Middle Name
	Last Name (B-LN)
	Birthdate (B-DOB)
	Birth Order (BO) for multiple births (e.g., twins, triplets)
	Gender (Sex, female, male and unknown)
**Biological Mother**	First Name (M-FN)
	Middle Name
	Last Name (M-LN)
	Birthdate (M-DOB)
	Mother’s Phone number (TN) Mother’s street Address (ADD)
**The other parent**	Other parent’s Phone number (TN) Other parent’s Street Address (ADD)

Note: Middle names were not used in the initial and further matching analysis.

**Table 3 children-09-01916-t003:** Summary of the three hospital linkage scenarios.

Scenario	Details	Outcome
Scenario 1: Primarily using mother’s details to link the mother and baby	1st letter of mother first name and first 2 letters of mother surname and mother’s date of birth (dd/mm/ or yyyy). Any inpatient management (iPM, inpatient admission data) data would be joined where the Unit Record (UR) number from above criterion joins to IP admission UR number, and the Child’s birthdate is between the IP admission date -1 day and separation date, and it is a maternity episode. Additional Birth Outcomes System (BOS)/iPM data will join where the UR number from the 1st criterion joined BOS UR number.	The hospital identified 1919 potential pairs
Scenario 2: Primarily using baby’s details to link the mother and baby	1st two letters of baby surname OR Mother LN + baby (DOB). Join any inpatient episode through baby’s UR number In addition, BOS/iPM data were joined by their common UR number.	The hospital identified 2289 potential pairs
Scenario 3: Using all available variables for babies without information about their mothers	Using babies’ FN and LN, DOB Using baby’s PII and obtained baby’s other parent’s TN and ADD.	The hospital identified 53 out of 58 babies

Note: iPM = Patient Administration System; BOS = Outcomes System; FN = first name; LN = last name; DOB = date of birth; BO = birth order; TN = telephone; ADD = addresses.

## Data Availability

We are not able to share the participant data for this article due to privacy and legislation requirements. For matching algorithms, please contact Y.J.H.

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
