# Peer review of "Improving Cohort-Hospital Matching Accuracy through Standardization and Validation of Participant Identifiable Information"

_children, 2022, doi:10.3390/children9121916_

Round 1

Reviewer 1 Report

The authors have addressed a meaningful and relevant topic that deserves full attention, especially with regard to increasing the development of accuracy within larger datasets. It could be of substantial benefit to other cohort studies linking multiple variable data.

The manuscript is clearly written. Methods used for the analyses are appropriate and adequately and extensively described.

General remarks:

My main concern, as the authors rightfully address in the discussion (lines 382-391) is the safety issue of the data. Indeed, my main concern remains linking information with ID numbers at the risk that confidential information may be made public. The authors describe in detail the security measures taken when they collected and managed the data as would be possible to achieve within the Australian health system. However, this may not be readily applicable to other settings. For example, in LMICs, only handwritten medical reports are often available and possibilities for data-linking are limited. It would be good to explicitly bring this up in the limitation section.

Comments

Table 2: baby’s surname: the child is born under the mother's name but is acknowledged by the father shortly after birth and will then bear the father's surname. Has that been taken into account?

Line 199: sex: was ‘unknown’ taken into account, for example in the case of children with ambiguous genitalia?

Lines 275-276: refers to table 3 describing details of all 10 steps and variables. However, these 10 steps are not clearly described: please clarify and/or number each step in the table

S-Figure 1 clearly describes important information and should not be supplemental

Minor comments

Line 236: typo: undertake

Line 260: 2 x the

Lines 224 and 457: written informed consent? Please add

Author Response

Reviewer1

The authors have addressed a meaningful and relevant topic that deserves full attention, especially with regard to increasing the development of accuracy within larger datasets. It could be of substantial benefit to other cohort studies linking multiple variable data.

Thanks, we also believe this will bring a wide range of interests for other cohort studies.

NA

The manuscript is clearly written. Methods used for the analyses are appropriate and adequately and extensively described.

Thanks.

NA

My main concern, as the authors rightfully address in the discussion (lines 382-391) is the safety issue of the data. Indeed, my main concern remains linking information with ID numbers at the risk that confidential information may be made public. The authors describe in detail the security measures taken when they collected and managed the data as would be possible to achieve within the Australian health system. However, this may not be readily applicable to other settings. For example, in LMICs, only handwritten medical reports are often available and possibilities for data-linking are limited. It would be good to explicitly bring this up in the limitation section.

Thank you.  Our first limitation now states: Therefore, our findings are likely to apply for birthing hospital-to-cohort exchanges throughout Victoria and potentially nationally and to other countries with computerized administrative databases; however, we acknowledge that these may not always be available in low-middle income countries, which may raise different security concerns than managed here.

Page 12 Line 443-447

Table 2: baby’s surname: the child is born under the mother's name but is acknowledged by the father shortly after birth and will then bear the father's surname. Has that been taken into account?

We have added this information under the discussion section of ‘implementations’ to read “Many Australian babies are born under mothers’ name, but not all, Australian babies after birth (eg babies born to sole mothers, gay parents, or whose parents choose to give them the mother’s surname). Our approach has proved sufficient flexibility to match regardless of whether babies’ surname is the same or different from their mothers’ surnames.”

Page 12 Line 420-424

Line 199: sex: was ‘unknown’ taken into account, for example in the case of children with ambiguous genitalia?

SLK-581 allows to use three options to code sex: 1-Male,2-Female,9-Unknown, which was used in the analysis process. We have added the reference. In Table 2 we added “unknown” under baby “Gender (Sex: female, male and unknown).”

Page 7 line 209 Table 2

Lines 275-276: refers to table 3 describing details of all 10 steps and variables. However, these 10 steps are not clearly described: please clarify and/or number each step in the table

We now have referred to further details in Supplemental Table 6.

Page 9 line 307

Supplemental Table 6

S-Figure 1 clearly describes important information and should not be supplemental

Figure moved to the Results section and renamed as Figure 3.

Page 8 Line 278-281

Line 236: typo: undertake

Corrected

Page 7 Line 251

Line 260: 2 x the

Corrected

Page 9 Line 292

Lines 224 and 457: written informed consent? Please add

Yes, we have added the written information consent. It reads “Written informed consent was obtained from participants.”

Page 7 Line 239; Page 19 line 529

Reviewer 2 Report

Dear authors your work is interesting.

Please review your paper to make sure that it is within the interests of the journal readers and it is aligned with the journal's aims and scope.

My other comments relate to the clarity of the methodology and those manual checks mentioned in the paper that are required to improve the accuracy of your approach. Would manual checks be the appropriate approach to checking of perhaps thousands of data records?

Finally, a few minor spell-checking typos to paly attention to:

63 (5].),164 (..), 176/183 (Figure 1 – Figure, 2), 269 (numbers without “,”), 310 (1819)before -space),418 (Ee), 420(ipm - iPM), 426 (2nd line after the 3rd line), 429 (numbers without “,”), 531 ():), 532 ():)

Author Response

Reviewer2

Please review your paper to make sure that it is within the interests of the journal readers, and it is aligned with the journal's aims and scope.

We believe this study fits the interests of the journal readers and is aligned with Children’s aim and scope, specifically Public Health and Epidemiology. It proposes an effective method to link cohort and birthing hospitals data for children and their mothers, which will enable wide-ranging perinatal research to improve children’s health outcomes.

No change

My other comments relate to the clarity of the methodology and those manual checks mentioned in the paper that are required to improve the accuracy of your approach. Would manual checks be the appropriate approach to checking of perhaps thousands of data records?

We agree that in this pilot study, manual checks acted as the essential ‘gold standard’ to determine the rates of true matches and hence the most optimised matching approach for scale up to >50 hospitals. For checking this pilot with 1819 mother babies’ pairs were appropriate. However, it will become a big challenge for >50 hospitals unless we have an optimised approach. Through the pilot, SLK-5881 approach reached 97%, then only left 3% (55) for the manual check for mother -baby pairs. Please see edits to the methods clarifying “We used manual comparison as the gold standard to confirm all true matches for this pilot.”

Page 7 Line 232-233

Finally, a few minor spell-checking typos to paly attention to: 63 (5].),164 (..), 176/183 (Figure 1 – Figure, 2), 269 (numbers without “,”), 310 (1819)before -space),418 (Ee), 420(ipm - iPM), 426 (2nd line after the 3rd line), 429 (numbers without “,”), 531 ():), 532 ():)

Thanks, all revised.

Page 2 line 63;

Page 6 lines 171; Page 6 line 190;

Page 9, lines 301;

Page 13, 481;

page 14, ipm-iPM;

s-Figure 1 edited and moved to main text as figure 3.

Page 17, line 495;

Page 21, line 605, 606;

Round 2

Reviewer 2 Report

Very interesting research. It could be improved if you could elaborate more on the need of any manual tasks you describe in your paper.